# Motor-Independent Cognitive Testing in Motor Degenerative Diseases

**DOI:** 10.3390/jcm11030814

**Published:** 2022-02-03

**Authors:** Henning Schmitz-Peiffer, Elisa Aust, Katharina Linse, Wolfgang Rueger, Markus Joos, Matthias Löhle, Alexander Storch, Andreas Hermann

**Affiliations:** 1Department of Neurology, Technische Universität Dresden, 01307 Dresden, Germany; henning.schmitz-peiffer@uniklinikum-dresden.de (H.S.-P.); elisa.aust@uniklinikum-dresden.de (E.A.); katharina.linse@uniklinikum-dresden.de (K.L.); 2Deutsches Zentrum für Neurodegenerative Erkrankungen (DZNE) Dresden, 01307 Dresden, Germany; 3Interactive Minds Research, Interactive Minds Dresden GmbH, 01309 Dresden, Germany; rueger@interactive-minds.com (W.R.); joos@interactive-minds.com (M.J.); 4Department of Neurology, University of Rostock, 18051 Rostock, Germany; matthias.loehle@med.uni-rostock.de (M.L.); alexander.storch@med.uni-rostock.de (A.S.); 5Deutsches Zentrum für Neurodegenerative Erkrankungen (DZNE) Rostock/Greifswald, 18147 Rostock, Germany; 6Center for Transdisciplinary Neurosciences Rostock (CTNR), University Medical Center Rostock, University of Rostock, 18147 Rostock, Germany; 7Translational Neurodegeneration Section “Albrecht-Kossel”, Department of Neurology, University Medical Center Rostock, University of Rostock, 18147 Rostock, Germany

**Keywords:** amyotrophic lateral sclerosis, eye tracking, cognition, executive functions, neuromuscular diseases, neuropsychological tests, Parkinson’s disease, Trail Making Test

## Abstract

Cognitive function is tested through speech- or writing-based neuropsychological instruments. The application and validity of those tests is impeded for patients with diseases that affect speech and hand motor skills. We therefore developed a “motor-free” gaze-controlled version of the Trail Making Test (TMT), including a calibration task to assess gaze accuracy, for completion by means of an eye-tracking computer system (ETCS). This electronic TMT version (eTMT) was evaluated for two paradigmatic “motor-neurodegenerative” diseases, Parkinson’s disease (PD) and amyotrophic lateral sclerosis (ALS). We screened 146 subjects, of whom 44 were excluded, e.g., because of vision deficits. Patients were dichotomized into subgroups with less (ALS−, PD−) or severe motor affection (ALS+, PD+). All 66 patients and all 36 healthy controls (HC) completed the eTMT. Patients with sufficient hand motor control (ALS−, PD−, PD+) and all HC additionally completed the original paper–pencil-based version of the TMT. Sufficient and comparable gaze fixation accuracy across all groups and the correlations of the eTMT results with the TMT results supported the reliability and validity of the eTMT. PD+ patients made significantly more errors than HC in the eTMT-B. We hereby proved the good applicability of a motor-free cognitive test. Error rates could be a particularly sensitive marker of executive dysfunction.

## 1. Introduction

The assessment of cognitive functions can be problematic in patients with neurological disorders due to disease-related motor impairment [1,2]. Neuropsychological tests are a key component of the objective and reliable evaluation of neurocognitive functioning, but they require (unimpaired) speech or motor skills. Patients with Parkinson’s disease (PD) or amyotrophic lateral sclerosis (ALS), for example, show a severe deterioration of speech, hand, and finger motor performance in the course of their disease, making neuropsychological testing impossible or leading to a poorer performance—particularly for speed/reaction time-dependent tests. Since these effects of motor impairment cannot be delimited from the effects of cognitive impairment on a patient´s test performance, results cannot be assumed as valid. Data on cognitive dysfunction in progressed disease stages are thus extremely limited [2,3,4]. So far, only a few studies have addressed these limitations of application and validity [5,6]. How can technology help us to overcome them? A promising approach is the development of eye-tracking-based neuropsychological tests. Since eye movement control is preserved in general or at least for a very long time in the ALS disease course, inter alia [7], such tests can help to overcome the confounding effects of motor impairment and the resulting lack of diagnostic value in cognitive assessment [8,9]. For very advanced ALS-patients in an incomplete locked-in stage (LIS), it is the only remaining access to cognitive function.

We have developed a digital version of the Trail Making Test (TMT) which is applied by means of an eye-tracking computer system (ETCS). The TMT is a widely used paper–pencil-based tool for the assessment of visual processing speed and executive functioning [10]. In contrast to some previously published ETCS versions of the TMT [11,12], our solution is exactly based on the original template without horizontal flipping or rotation of the items.

The purpose of our present study was to evaluate the usability and sensitivity of this electronic eye-tracking-based “motor-free” TMT-version (eTMT), which can be adapted to patients with different extents of motor impairment by using a fixed or a mobile and adjustable ETCS. Therefore, we examined patients with the two paradigmatic “motor” neurodegenerative diseases PD and ALS in early to advanced disease stages, also taking oculomotor functioning into account.

## 2. Materials and Methods

Patients and healthy controls (HC) were prospectively recruited from a specialist outpatient clinic at the University Hospital Carl Gustav Carus Dresden, Germany and from a patient network (ALS-mobil eV, r. a.). Patients with PD had an established diagnosis according to MDS clinical diagnostic criteria [13]. ALS diagnosis was determined according to the revised El Escorial criteria [14]. Disease severity was rated using the ALS-functional rating scale revised (ALSFRS-R) for ALS [15] and the Hoehn and Yahr score for PD [16]. Exclusion criteria comprised atypical PD syndromes, aphasia, and visual field defects or other optical or optomechanical limitations. Moreover, patients were not included in the study if they had a previously established diagnosis of any psychiatric disease, in particular frontotemporal dementia or PD dementia as defined by ICD-10 criteria (F00–F03).

ALS patients in LIS (in the ALS+ group) accustomed themselves to ETCS use as a communication device in daily life for at least two weeks before assessment.

For all gaze data collection and test procedures, a monocular Eyegaze Edge remote IR eye-tracking device (LC Technologies, Inc., South Plainfield, NJ, USA) with a sampling rate of 60 Hz was used, attached to the bottom of the stimulus screen. The ETCS screen was positioned fronto-parallel at a distance of 60–70 cm from the subject´s face. Patients with sufficient body control executed the tests seated and with the head rested upon a chin rest. All other patients were tested with a mobile and adjustable ETCS and either lying in bed or seated in a wheelchair (see Appendix A).

Before completing the eTMT, participants had to pass a 13-point calibration task, which secured the minimum required fixation accuracy for the gaze-controlled operation of the ETCS in all four quadrants and thus all visual directions. Failure or trouble in this calibration task led to the cessation of study procedures.

Fixation or gaze accuracy in the calibration task was analyzed as the average angular degree of deviance (°) between the target and gaze fixation points (see Appendix A).

Electronic gaze-controlled versions of both subtests of the TMT (eTMT-A, eTMT-B) were built, according to the original test form with regard to the layout of the test templates. As in the original test, they included exercise tasks preceding the respective subtest, each consisting of the first eight items of eTMT-A and eTMT-B (see Appendix A). The eTMT was written in Object Pascal and implemented in and executed from the NYAN^®^ eye-tracking data analysis suite (Interactive Minds Dresden GmbH, Dresden, Germany). Selection of an item required a minimal fixation duration of 100 ms (dwell delay 50 ms, dwell time 50 ms) with an error time of 550 ms (dwell delay 50 ms, dwell time 500 ms). The dwell delay and dwell time scores were chosen according to previously published recommendations [17,18]. Visual selection of an item by exceeding those time thresholds was counted as intended fixation. Each correct selection of an item was followed by a black line connecting the correctly selected and the previous item. Wrongly selected items were counted as errors. Cessation of the respective test was enforced manually by the test administrator when time limits were reached: 180 s for eTMT-A and 300 s for eTMT-B in accordance with the TMT test protocol. The evaluated parameters of the eTMT were the completion times for eTMT-A and eTMT-B, the ratio of the completion times of the two subtests (eTMT-B/eTMT-A), and the number of errors as well as the ratio of errors (eTMT-B/eTMT-A). As additional parameters, the total number of fixations, the number of fixations per second, and the number of errors per fixation were collected and analyzed.

Patients with sufficient motor skills and all HC completed both the TMT (paper–pencil version) and eTMT in random order. Patients with severe motor impairment only completed the eTMT. For the written TMT, the completion times for both subtests and the ratio of the completion times were evaluated.

SPSS Software (version 25; IBM, Inc., Chicago, IL, USA) was used for statistical analysis. Group comparisons for categorical data were conducted using a chi-squared test. The normal distribution of the eTMT data was tested by a Kolmogorov–Smirnov test and the homogeneity of variance was tested by Levene’s test. For the between-group comparison of the eTMT-variables, a Kruskal–Wallis test instead of ANOVA was used if one or both assumptions were violated. Adequate post hoc procedures were performed as indicated and adjusted for multiple pairwise comparisons. Correlations between TMT and eTMT were calculated with Spearman’s rho. Significance level was set at α = 5%.

## 3. Results

### 3.1. Sociodemographic and Clinical Data

We screened 146 participants (105 patients and 41 HC) for eligibility, *n* = 102 participants (66 patients, 36 HC) completed the study procedure (Figure 1). As presented in Figure 1, reasons for exclusion comprised visual deficits, disabling tremor, exhaustion, post hoc diagnosis of dementia, death prior to testing, or the defective calibration of ETCS. Bimodal distributions of ALSFRS-R scores in ALS patients (*p* < 0.001) and Hoehn and Yahr stages in PD patients (*p* < 0.001) allowed us to categorize patients into two groups with respect to disease severity. ALS patients were grouped into patients with higher motor impairment (ALS+) and lower motor impairment (ALS−), depending on whether their ALSFRS-R-score was below/equal to or higher than 20, respectively. PD patients were categorized into patients with higher motor impairment (PD+) and lower motor impairment (PD−), depending on whether they reached a Hoehn and Yahr score equal to/higher than 2.5 or below 2.5, respectively. While ALS was sporadic for all patients in the ALS+ group, three patients in the ALS− group had a family history of ALS. Genetic testing of those familial ALS cases confirmed one case of genetic ALS with C9ORF72, while for the other two genetic testing revealed no mutation in known ALS genes. Table 1 shows the most relevant characteristics for the final sample of ALS and PD patients.

### 3.2. Gaze Accuracy

No significant group differences were found in degrees of deviance between targets and gaze fixation (*F* (4, 97) = 1.44, *p* = 0.23) and thus for gaze accuracy in the calibration task (Table 2, Figure 2a).

### 3.3. Validity of eTMT: Correlation between eTMT and Written TMT

In total, 90 of the 102 enrolled participants were able to complete the written TMT, including only 2 of the ALS+ patients. The ALS+ group was therefore excluded from the respective analysis. Spearman’s rho revealed significant moderate correlations between TMT-A and eTMT-A (Figure 2b) as well as between TMT-B and eTMT-B (Figure 2c) for the HC group ((e)TMT-A: *r* = 0.433, *p* = 0.008; (e)TMT-B: *r* = 0.641, *p* < 0.001), the ALS- group ((e)TMT-A: *r* = 0.551, *p* = 0.009; (e)TMT-B: *r* = 0.526, *p* = 0.025), and the PD− group ((e)TMT-A: *r* = 0.551, *p* = 0.018; (e)TMT-B: *r* = 0.574, *p* = 0.010). No significant correlations between the test versions were found for the PD+-group ((e)TMT-A: *p* = 0.141; (e)TMT-B: *p =* 0.120).

### 3.4. Group Comparison for Performance in Written TMT

As presented in Table 2, there were no significant differences between the four subgroups for the completion speed in TMT-A (*F* (3, 86) = 1.78, *p* = 0.16), TMT-B (*F* (3, 86) = 2.21, *p* = 0.09), or for the TMT-B/A ratio (*F* (3, 86) = 0.37, *p* = 0.78).

### 3.5. Group Comparison for Performance in eTMT

In comparison to all other groups, PD+ patients showed the longest completion times for eTMT-A and eTMT-B, even though the Kruskal–Wallis test revealed no significant group difference. Groups did not differ significantly for the B/A ratio either (Table 2, Figure 2d,e). Regarding the number of errors, the groups did not differ significantly for eTMT-A but did for eTMT-B (H (4) = 9.786, *p* = 0.044). Post hoc pairwise comparisons showed that PD+ patients made significantly more errors than HC in the eTMT-B (*p* = 0.046; Figure 2f). Groups did not differ for the number of fixations in eTMT-A and eTMT-B.

Significant differences could be demonstrated for the number of fixations per second in eTMT-A (*t* (39.17) = 4.66, *p* = 0.003, ANOVA) and eTMT-B (H (4) = 21.31; *p* < 0.001; Kruskal–Wallis) (Figure 2g). Post hoc pairwise comparisons for eTMT-A pointed out that ALS+ patients made significantly fewer fixations per second compared to ALS- (0.51, 95% CI (0.003, 0.099); *p* = 0.006), to PD− (0.62, 95% CI (1.09, 0.13), *p* < 0.01), to PD+ (0.61, 95% CI (1.10, 0.12); *p* = 0.008), and to HC (0.71, 95% CI (1.14, 0.27), *p* < 0.001). For eTMT-B, Dunn–Bonferroni post hoc pairwise comparisons revealed significantly fewer fixations per second in the ALS+ subgroup than in HC (−41.36, *p* < 0.001) and in PD− (−34.47, *p* = 0.016). We found significant group differences for errors per fixation in eTMT-A (H (4) = 9.61, *p* = 0.047) and eTMT-B (H (4) = 11.53, *p* = 0.021; both Kruskal–Wallis) (Figure 2h). However, pairwise comparisons identified no significant differences between any groups for eTMT-A. In eTMT-B, however, PD+ patients made more errors per fixation compared to HC (24.5; *p* = 0.048).

## 4. Discussion

The present study demonstrates that the ETCS-based assessment of cognitive functioning in motor-restricted patients via eTMT is feasible in a standard clinical setting, even in patients lacking experience with ETCS. In contrast to previous studies, a mobile and adaptable ETCS that could be used without a chin rest allowed for the examination of patients with very severe motor limitations—which is the subpopulation of patients for whom eye-tracking-based solutions for cognitive testing are most relevant and needed [1,19]. Moreover, our eTMT template—unlike earlier versions—corresponds exactly to the template of the original paper–pencil-based TMT template. Moderate correlations between performance in eTMT and TMT for both subtests in all groups except PD+ are an important indicator for the validity of the developed eye-tracking version. Regarding the comparable fixation accuracy and thus oculomotor functioning between the HC and patient groups, we provide a tool for motor-independent cognitive testing from early to late disease stages and even in the LIS.

Looking at the classic TMT parameters, however, it seems surprising that ALS patients did not perform significantly worse than HC in the present study, which is also contradictory to previous eTMT results [12]. However, we investigated a group of very early ALS patients, while our group of advanced ALS patients (including LIS patients) showed a poorer, though not significantly worse, performance as measured by the key parameters of completion times and their ratio, but also by the parameter of errors. Regarding this, it also needs to be considered that LIS patients in this group of ALS+ patients used an ETCS as a communication aid and were therefore familiar with and practiced in dwell-time-based gaze control, which probably improved their performance in the eTMT. This practice effect could be reflected in the lower number of fixations per second made by the advanced ALS patients compared to all other groups. Moreover, a particularly strong positive selection bias for those very advanced patients can be assumed (e.g., of patients with good cognitive state and no problems with ETCS use), regarding the severity of motor impairment in this LIS of ALS.

Most unexpected, however, was that PD patients did not perform significantly poorer than HC regarding the primary TMT outcomes of completion times and their ratio, either in TMT or eTMT, which is in contrast to a number of previous findings showing progressive cognitive decline in PD [20,21]. However, the descriptive analysis of completion time at least points to a progression-related deceleration of cognitive, particularly executive, functioning in PD. The small sample size and therefore statistical power is one important explanation for a lack of statistical significance for those differences and is a main limitation of this study. Regarding the present data, another possible explanation is the presence of a stronger speed accuracy trade-off in the HC group, suggesting they tried harder to avoid mistakes with the consequence of a slower completion of the task.

The finding of increased error rates of PD+ patients in eTMT-B, however, indicates the deterioration of cognitive control or executive functioning, which might be related to altered diffusion in the frontal lobe activity, in line with the development of fronto-temporal dysfunction in a relevant proportion of patients with PD [21,22]. This result supports that errors made in the TMT serve as an important, sensitive marker for cognitive impairment [23,24]. Thereby, it emphasizes the value of error analysis, which is automated and therefore much easier and more reliable by means of the digital eTMT than the original paper–pencil-based one.

Severe ocular dysfunction and dyskinesia/tremor are major limitations of the presented technology. Moreover, we screened a relevant number of particularly PD patients who were too motorically impaired to perform the eTMT sitting in front of the ECTS and still too mobile (too much head movement) to complete it using the mobile ETCS version without a chin rest. Head-mounted systems might overcome these restrictions and even increase the usability of the approach [19]. The lack of a significant correlation between eTMT and TMT completion times only in the PD+ group could, moreover, hint to a lower validity of this test for these advanced patients. However, it is a likely alternative explanation that severe motor deficits in this patient group have a great but also heterogonous impact on TMT results as confounding factors—emphasizing the importance of “motor-free” cognitive testing.

Beside its small sample size, a main limitation of the study is a training effect for the gaze-controlled completion of the eTMT from part A to part B, particularly for ETCS-inexperienced subjects. This effect is strongly indicated by the comparison of the B/A ratio between eTMT and TMT: HC and PD patients need more time to complete eTMT-A than the TMT-A, while they present shorter completion times for eTMT-B than for TMT-B, and consequently lower B/−A ratios for the eTMT than for the TMT. This leads to the assumption that eTMT completion per se requires less time than TMT completion (gaze movements are quicker than hand movements), but that subjects lose time in eTMT-A for understanding and getting used to the dwell-time-based interaction. An important measure to solve or reduce this problem would be a more intensive ETCS training prior to the completion of eTMT. This could include, e.g., a writing task (for getting accustomed to the dwell-based gaze interaction in general) and more importantly a longer TMT-specific practice task. This would allow participants to understand and to get used to the principles and the interaction speed of the unfamiliar task. This eTMT practice might consist of, e.g., writing a particular word by connecting letters or connecting colored circles in a given order, according to the principles and design of eTMT (but with a totally different arrangement of items than in the actual task). Even then, however, it would be an important measure to collect normative data for the gaze-controlled eTMT, which is underlined by the comparison of TMT and eTMT data in the present study.

In summary, we provide first evidence for the usability of a new motor-independent and valid cognitive assessment instrument using ETCS for classical movement disorders. Conclusions from this pilot study’s results are limited by its small sample size and the training effect discussed above. Another limitation with regard to the written TMT is the lack of availability of fine motor or particularly hand/arm motor function scores for our cohort, which thus cannot be taken into account as an influencing factor on performance in this original test version. However, and importantly, the bimodal grouping and the comparison of patients based on that was not meant to evaluate the influence of hand function, but to investigate whether overall motor disease progression is associated with progressed cognitive impairment, and whether patients with overall very severe motor impairment can still be properly assessed using the gaze-controlled version of the TMT. Therefore, the main outcome for all patients was their performance in the eTMT. The analysis of the correlation between the written and eTMT was additionally used to evaluate the validity of the latter. Nevertheless, the in-depth analysis of the effect of dominant hand/arm function on written TMT test results would be interesting to follow up in a larger cohort. Further research is needed to verify the findings by examining cognitive functioning assessed by the eTMT—improved as discussed above—in a larger sample of early-stage to late-stage ALS and especially PD. Ideally, future studies would have a longitudinal design and assess additional neurocognitive functions, including specific oculomotor functions like reflexive pro-saccade latencies. What would also be of interest are correlations between performance in eTMT with anti-saccade-error-rates as another indicator of executive dysfunction, e.g., ref. [25,26] which can be assessed by means of ETCS.

## Figures and Tables

**Figure 1 jcm-11-00814-f001:**
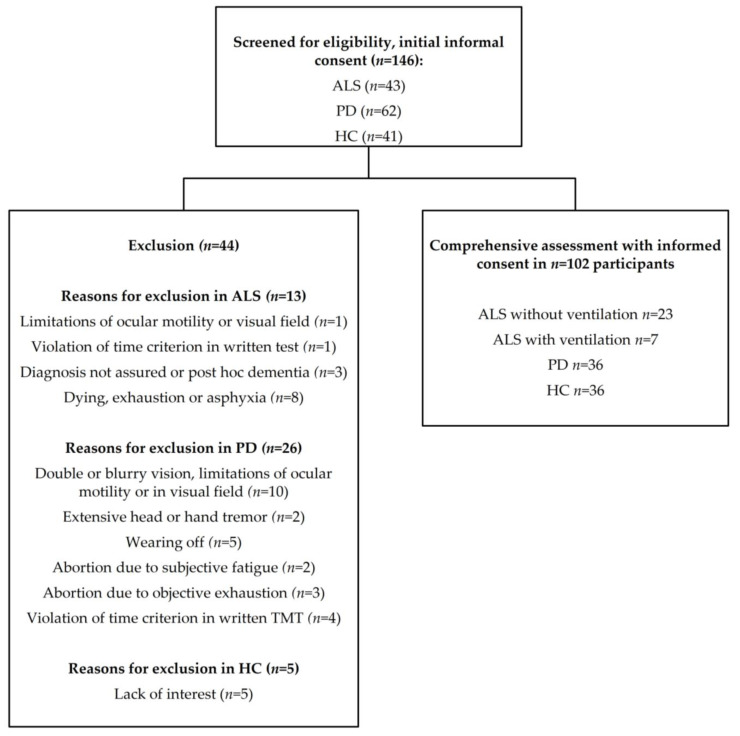
Flowchart of study enrollment for patients and healthy controls (HC).

**Figure 2 jcm-11-00814-f002:**
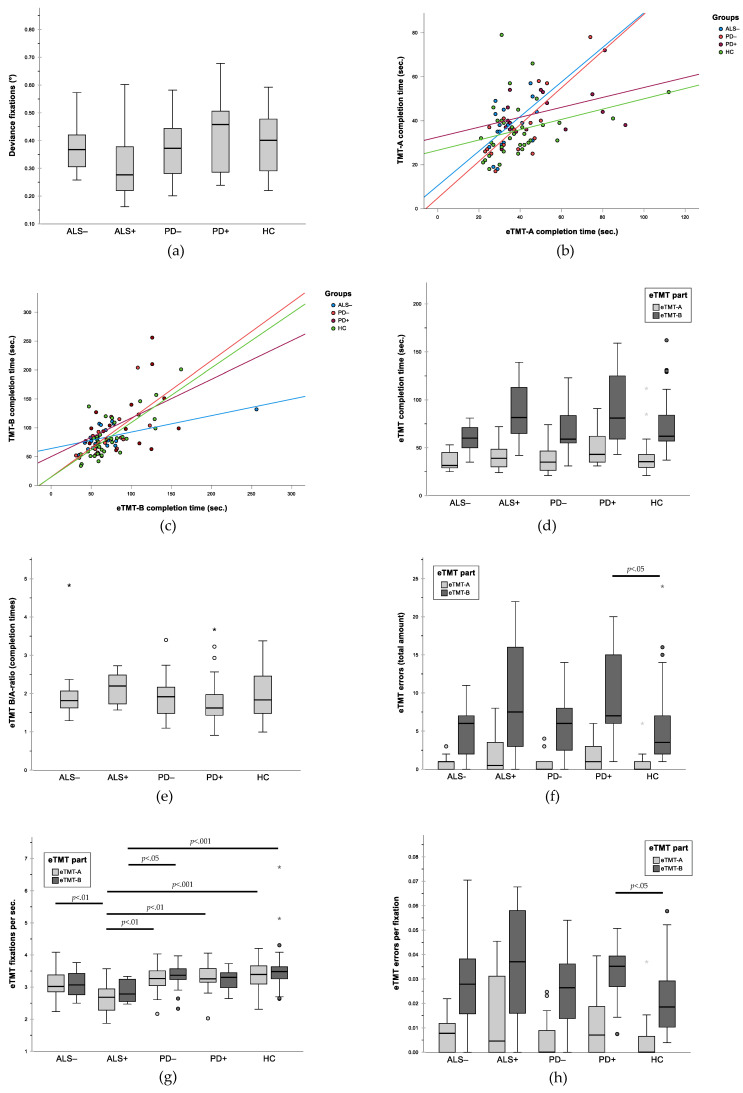
Group comparisons and correlations for gaze accuracy and performance in TMT and eTMT. (**a**) Deviance (in degrees) between target and fixation in calibration task as measure of gaze accuracy; (**b**) correlation between written TMT-A and eTMT-A; (**c**) correlation between written TMT-B and eTMT-B; (**d**) completion times for eTMT subtests in comparison between groups; (**e**) B/A ratio of completion times for eTMT subtests in comparison between groups; (**f**) total number of errors in eTMT subtests in comparison between groups; (**g**) fixations per second in eTMT subtests in comparison between groups; (**h**) errors per fixation in eTMT subtests in comparison between groups. In figures (**d**–**h**): ° outliers: values below the first quartile or above the third quartile with a distance of 1.5 to 3 times the interquartile range; * extreme outliers: values below the first quartile or above the third quartile with a distance of more than 3 times the interquartile range.

**Table 1 jcm-11-00814-t001:** Demographic and clinical characteristics and psychosocial outcomes.

Characteristic	ALS− (*n* = 18)	ALS+ (*n* = 12)	PD− (*n* = 19)	PD+ (*n* = 17)	HC (*n* = 36)	Total (*n* = 102)	*p* Value ^1^
Age, years	60.2 ± 9.4	56.3 ± 8.5	61.7 ± 9.1	65.6 ± 8.8	62.3 ± 8.8	61.7 ± 9.1	n. s.
Sex, male	61.1%	50.0%	63.2%	76.5%	50.0%	58.8%	n. s.
Education, years	13.4 ± 2.2	15.4 ± 2.7	14.8 ± 2.3	14.9 ± 3.2	14.2 ± 3.1	14.4 ± 2.8	n. s.
Right-handedness	88.9%	100%	100%	100%	91.7%	97%	n. s.
Employed	16.7%	0%	26.3%	0%	44.4%	23.5%	<0.001
Disease duration (years)	2.9 ± 2.8	6.2 ± 3.7	7.4 ± 5.5	13.2 ± 6.4	-	-	<0.001
ALS-type, sporadic	83.3%	100%	-	-	-	-	n. s.
ALS-onset, spinal:bulbar	50%:50%	75%:25%	-	-	-	-	n. s.
ALS-ventilated	0%	58.3%	-	-	-	-	<0.001
ALSFRS-R score ^a^	36.4 ± 5.7	5.8 ± 6.0	-	-	-	-	<0.001
PD-equivalent	-	-	31.6%	47.1%	-	-	n. s
PD-hypo-kinetic-rigid	-	-	42.1%	41.2%	-	-	n. s.
PD-tremor dominant	-	-	26.3%	11.8%	-	-	n. s.
H&Y score ^b^: Median (Range)	-	-	2.0 (1.5 – 2.5)	3.0 (3.0 – 4.0)	-	-	<0.001
UPDRS III-score ^c^	-	-	18.1 ± 8.5	27.9 ± 7.2	-	-	<0.01
DBS (%)	-	-	10.5%	52.9%	-	-	<0.01

Data presented as Mean ± SD (if not stated otherwise). Abbreviations: ALS, Amyotrophic lateral sclerosis; PD, Parkinson’s disease; ALSFRS, ALS-Functional Rating Scale; H&Y, Hoehn and Yahr; ALS+, ALS patients with severe motor impairment (ALSFRS-R < 20); ALS−, ALS patients with less motor impairment (ALSFRS-R > 20); PD+, PD patients with severe motor impairment (H&Y < 2.5); PD-, PD patients with less motor impairment (H&Y > 2.5); HC, healthy controls; UPDRS, Unified Parkinson’s Disease Rating Scale; DBS, deep brain stimulation. ^a^ Scores on the ALSFRS-R range from 0 to 48, with lower scores denoting a worse condition; ^b^ scores on the H&Y range from 0 to 5, with higher scores denoting a worse condition; ^c^ scores on the UPDRS III range from 0 to 108, with higher scores denoting more disability. ^1^ ANOVA or χ^2^ test or Mann–Whitney U-Test.

**Table 2 jcm-11-00814-t002:** Results for TMT and eTMT.

Written TMT							
	ALS− (*n* = 18)	(ALS+) † (*n* = 2)	PD− (*n* = 19)	PD+ (*n* = 17)	HC (*n* = 36)	Total (*n* = 90)	*p* Value
Completion time TMT-A	38.1 ± 11.4	(70.0 ± 8.5) †	36.2 ± 14.4	43.9 ± 10.7	35.7 ±12.9	38.6 ±13.5	0.157 ^1^
Completion time TMT-B	84.5 ± 20.7	(109.0 ± 53.7) †	83.3 ±36.0	110.1 ± 53.6	83.8 ± 36.6	89.3 ± 38.8	0.093 ^1^
Ratio TMT-B/-A	2.3 ± 0.7	(1.5 ± 0.6) †	2.4 ± 0.7	2.7 ± 1.5	2.4 ± 0.8	2.4 ± 0.9	0.777 ^1^
**Electronic TMT (eTMT)**							
	**ALS−** **(*n* = 18)**	**ALS+** **(*n* = 12)**	**PD−** **(*n* = 19)**	**PD+** **(*n* = 17)**	**HC** **(*n* = 36)**	**Total** **(*n* = 102)**	***p* Value**
Fixation deviance (°)	0.374 ± 0.078	0.325 ± 0.142	0.375 ± 0.108	0.423 ± 0.135	0.392 ± 0.108	0.383 ± 114	0.228 ^1^
Completion time eTMT-A	35.22 ± 8.71	42.00 ± 15.71	37.53 ± 13.55	50.65 ± 19.91	39.39 ±17.61	40.49 ± 16.35	0.061 ^2^
Completion time eTMT-B	71.61 ± 47.99	88.33 ± 29.68	67.95 ± 23.28	89.65 ± 36.07	73.22 ± 29.44	76.47 ± 33.97	0.076 ^2^
Ratio completion time eTMT-B/A	1.96 ± 0.77	2.15 ± 0.40	1.89 ± 0.58	1.88 ± 0.78	1.95 ± 0.58	1.95 ± 0.63	0.399 ^2^
Amount of errors eTMT-A	0.83 ± 0.92	1.92 ± 2.81	0.89 ± 1.33	2.00 ± 2.12	0.64 ± 1.42	1.10 ± 1.74	0.085 ^2^
Amount of errors eTMT-B	8.17 ± 12.58	9.67 ± 7.34	5.58 ± 3.79	9.76 ± 5.76 ^a^	6.00 ± 6.09	7.36 ± 7.50	<0.05 ^2^
Ratio errors eTMT-B/A	7.43 ± 7.87	6.22 ± 6.06	4.29 ± 4.69	4.05 ± 4.96	6.19 ± 6.51	5.61 ± 6.04	0.199 ^2^
eTMT-A: Fixations total amount	112.7 ± 36.9	110.4 ± 40.6	124.5 ± 51.3	163.1 ± 63.1	132.8 ± 62.0	130.1 ± 56.0	0.055 ^2^
eTMT-B: Fixations total amount	223.0 ± 154.8	246.4 ± 67.7	226.0 ± 81.5	289.7 ± 120.3	271.2 ± 131.9	254.4 ± 121.2	0.135 ^2^
Fixations per sec. eTMT-A	3.2 ± 0.5 ^b^	2.7 ± 0.5 ^a^	3.3 ± 0.4 ^b^	3.3 ± 0.4 ^b^	3.4 ± 0.5 ^b^	3.2 ± 0.5	<0.001 ^1^
Fixations per sec. eTMT-B	3.1 ± 0.4	2.9 ± 0.3 ^a^	3.3 ± 0.4 ^b^	3.2 ± 0.3	3.7 ± 1.3 ^b^	3.4 ± 0.9	<0.001 ^2^
Errors per fixation eTMT-A	0.007 ± 0.007	0.015 ± 0.018	0.006 ± 0.008	0.012 ± 0.013	0.003 ± 0.007	0.007 ± 0.01	<0.05 ^2^
Errors per fixation eTMT-B	0.030 ± 0.018	0.037 ± 0.023	0.024 ± 0.015	0.032 ± 0.011^a^	0.021 ± 0.014	0.027 ± 0.017	<0.05 ^2^

Data presented as Mean ± SD (if not stated otherwise). Abbreviations: TMT, Trail Making Test; eTMT, electronic TMT; ALS, amyotrophic lateral sclerosis; PD, Parkinson’s disease; ALS+, ALS patients with severe motor impairment (ALSFRS-R < 20); ALS−, ALS patients with less motor impairment (ALSFRS-R > 20); PD+, PD patients with severe motor impairment (H&Y < 2.5); PD−, PD patients with less motor impairment (H&Y > 2.5); HC, healthy controls. † Data were excluded from calculation. ^1^ Univariate ANOVA; ^2^ Kruskal–Wallis test. ^a^ Post hoc significant difference compared to HC (*p* < 0.05; Dun–Bonferroni or Games–Howell test); ^b^ significant difference compared to ALS− (*p* < 0.05; Dun–Bonferroni or Bonferroni adjusted test).

## Data Availability

Data are available upon request.

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
