# Peer review of "Motor-Independent Cognitive Testing in Motor Degenerative Diseases"

_jcm, 2022, doi:10.3390/jcm11030814_

Round 1

Reviewer 1 Report

1) In the results the authors discuss that they "categorize patients into two groups with respect to disease severity. ALS patients were grouped into patients with higher motor impairment
(ALS+) and lower motor impairment (ALS-), depending on whether their ALSFRS-R-score
was below/equal or higher than 20, respectively." _ The problem with this is that since the normal trail making test mostly relies on hand function- one could imagine a situation where a ALS patient has perfectly normal dominant hand function but have lost points on the ALSFRS related to loss of ambulation, climbing steps, turning in bed, respiratory insufficiency - etc thus bringing there ALSFRS score to less than 20 while hand function remains nearly normal. This patient would be very different from a patient with ALS whose ALSFRS score is >20 but has only lost significant dominant hand function. Thus they might be characterized as less impaired but actually have more trouble with the test. A possibly better way to stratify the patients is by using the Fine motor score of the ALSFRS which more specifically addresses hand function. 

Author Response

Response: Thank you very much for pointing that out. It is true, that the overall ALSFRS-R score might not reflect the degree of paresis of the (dominant) hand, which is why even patients with a high ALSFRS-R total score can have significant problems in completing the written TMT due to hand motor impairment.

Unfortunately, data are anonymized and we do not have access to the ALSFRS-R-results at level of the single items, which prevents us from reporting the handwriting-score (item 4) or the fine motor subscore, respectively. We are very aware that this is a limitation and mention it as such in the revised manuscript.

However and importantly, the bimodal grouping and the comparison of patients based on that was not meant to evaluate the influence of hand function, but to investigate whether overall motor disease progression is associated with progressed cognitive impairment and whether patients with overall very severe motor impairment can still be properly assessed using the eye-gaze version of TMT.

Since it was clear that we could not investigate all patients with written TMT, the main outcome was the gaze-based eTMT version of the test in all patients. The analysis of the correlation between written and eTMT was additionally used to investigate validity of the latter. Nevertheless, this analysis should take hand motor function/handwriting or fine motor skills into account, which we are not able to provide. We would like to additionally note that we wouldn´t expect this analysis or inclusion of hand motor function to deliver very meaningful results. Considering that all patients in the ALS- group (less impaired) were able to complete the written TMT, there was no wide range/ a low variance of hand motor functions in our also rather small cohort.

Reviewer 2 Report

Peiffer et al.  present work to address a significant question in clinical ALS.  The authors are developing a means to address cognitive function despite significant motor decline.  They present the eye-gaze based TMT to allow for analysis in advanced and locked in patients.  The video in supplemental data is very helpful in understanding the process. The design and analysis are well thought out.

Several concerns:

  1. The results for ALSFRS reportedly separate in a bimodal grouping, but the data is not in the supplemental data.  Also, there is no ALSFRS subscore discussion. What was the range of hand function in the ALS+ group? What percentage are bulbar dysfunction?
  2. 83.3% of the ALS- are sporadic.  Was C9-related disease patients included?
  3. The sample size is small, and this testing requires training that is difficult to fully control for, but this is a pilot study.  It should be clearer that any conclusion on results is limited. 
  4. FTD and PD-dementia are excluded.  How is this defined?

Author Response

Response: We deeply thank the reviewer for this positive evaluation. The following are our answers to each of the helpful comments and questions:

Point 1: The results for ALSFRS reportedly separate in a bimodal grouping, but the data is not in the supplemental data.  Also, there is no ALSFRS subscore discussion. What was the range of hand function in the ALS+ group? What percentage are bulbar dysfunction?

Response 1:  Unfortunately, data are anonymized and we do not have access to the ALSFRS-R-results at level of the single items. This prevents us from reporting the handwriting-score (item 4) or the fine motor subscore as indicators of hand function as well as the (other) ALSFRS-subscores, respectively. We are very aware that this is a limitation and we mention as such in the revised manuscript.

Regarding hand motor function, we would like to note that however, all patients in the ALS- group (group with less motor impairment) were able to complete the written TMT; while this was true for only two patients in the ALS+ group, with most of those patients being in locked-in-state with no or only minimal residual motor function. Therefore, the range/variance of hand function was generally small in each group. However, we agree that it should have been reported and ideally taken in consideration for the analysis of the written TMT.

We added the percentage of patients with bulbar onset.

Point 2: 83.3% of the ALS- are sporadic. Was C9-related disease patients included?

Response 2:  We added this information to the manuscript.

Point 3: The sample size is small, and this testing requires training that is difficult to fully control for, but this is a pilot study.  It should be clearer that any conclusion on results is limited. 

Response 3: We attempted to make this clearer.

Point 4: FTD and PD-dementia are excluded. How is this defined?

Response 4: Any patients having preexisting diagnosis of FTD or PD dementia were excluded from the study, which we attempted to outline clearer in the revised version of the manuscript.